# And Yet It Moves: Oxidation of the Nuclear Autoantigen La/SS-B Is the Driving Force for Nucleo-Cytoplasmic Shuttling

**DOI:** 10.3390/ijms22189699

**Published:** 2021-09-08

**Authors:** Nicole Berndt, Claudia C. Bippes, Irene Michalk, Tabea Bartsch, Claudia Arndt, Edinson Puentes-Cala, Javier Andrés Soto, Liliana R. Loureiro, Alexandra Kegler, Dominik Bachmann, Joanne K. Gross, Tim Gross, Biji T. Kurien, R. Hal Scofield, A. Darise Farris, Judith A. James, Ralf Bergmann, Marc Schmitz, Anja Feldmann, Michael P. Bachmann

**Affiliations:** 1Department of Radioimmunology, Institute of Radiopharmaceutical Cancer Research, Helmholtz-Zentrum Dresden-Rossendorf (HZDR), 01328 Dresden, Germany; n.berndt@hzdr.de (N.B.); t.bartsch@hzdr.de (T.B.); c.arndt@hzdr.de (C.A.); epuentes@corrosion.uis.edu.co (E.P.-C.); jav.soto@mail.udes.edu.co (J.A.S.); l.loureiro@hzdr.de (L.R.L.); a.kegler@hzdr.de (A.K.); r.bergmann@hzdr.de (R.B.); a.feldmann@hzdr.de (A.F.); 2Institute of Immunology, Medical Faculty Carl Gustav Carus Dresden, Technische Universität Dresden, 01307 Dresden, Germany; claudia_bippes@gmx.de (C.C.B.); Irene.Michalk@uniklinikum-dresden.de (I.M.); marc.schmitz@tu-dresden.de (M.S.); 3Corporación para la Investigación de la Corrosión (CIC), Piedecuesta 681011, Colombia; 4Instituto de Investigación Masira, Facultad de Ciencias Médicas y de la Salud, Universidad de Santander, Cúcuta 540001, Colombia; 5Tumor Immunology, University Cancer Center (UCC), University Hospital Carl Gustav Carus Technische Universität Dresden, 01307 Dresden, Germany; domi_bachmann@hotmail.de; 6Arthritis and Clinical Immunology Program, Oklahoma Medical Research Foundation, University of Oklahoma Health Sciences Center, Oklahoma City, OK 73104, USA; Jody-Gross@omrf.org (J.K.G.); Tim-Gross@omrf.org (T.G.); Biji-Kurien@omrf.org (B.T.K.); hal-scofield@omrf.org (R.H.S.); Darise-Farris@omrf.org (A.D.F.); Judith-James@omrf.org (J.A.J.); 7Department of Biophysics and Radiobiology, Semmelweis University, 1094 Budapest, Hungary; 8National Center for Tumor Diseases (NCT), 03128 Dresden, Germany

**Keywords:** anti-La/SS-B antibodies, autoimmunity, La/SS-B autoantigen, systemic lupus erythematosus, primary Sjögren’s Syndrome, redox sensor, shuttling of La protein

## Abstract

Decades ago, we and many other groups showed a nucleo-cytoplasmic translocation of La protein in cultured cells. This shuttling of La protein was seen after UV irradiation, virus infections, hydrogen peroxide exposure and the Fenton reaction based on iron or copper ions. All of these conditions are somehow related to oxidative stress. Unfortunately, these harsh conditions could also cause an artificial release of La protein. Even until today, the shuttling and the cytoplasmic function of La/SS-B is controversially discussed. Moreover, the driving mechanism for the shuttling of La protein remains unclear. Recently, we showed that La protein undergoes redox-dependent conformational changes. Moreover, we developed anti-La monoclonal antibodies (anti-La mAbs), which are specific for either the reduced form of La protein or the oxidized form. Using these tools, here we show that redox-dependent conformational changes are the driving force for the shuttling of La protein. Moreover, we show that translocation of La protein to the cytoplasm can be triggered in a ligand/receptor-dependent manner under physiological conditions. We show that ligands of toll-like receptors lead to a redox-dependent shuttling of La protein. The shuttling of La protein depends on the redox status of the respective cell type. Endothelial cells are usually resistant to the shuttling of La protein, while dendritic cells are highly sensitive. However, the deprivation of intracellular reducing agents in endothelial cells makes endothelial cells sensitive to a redox-dependent shuttling of La protein.

## 1. Introduction

Precipitating antibodies in patients with Sjögren’s Syndrome were first described in 1958 by Jones [1] and further characterized by Anderson et al. in 1961 and 1962 [2,3]. These antibodies were “re-detected” at the end of the 1960s. In 1969 and 1974, Morris Reichlin’s team described the autoantibodies of cytoplasmic antigens in sera of patients with systemic lupus erythematosus (SLE) and termed them Ro and La antibodies (Abs) [4,5]. In 1975, Eng M. Tan and his team described autoantibodies to the Sjögrens’s syndrome-associated antigens A and B (SS-A/SS-B) [6]. Originally La and Ro were described as cytoplasmic antigens, while SS-A and SS-B were found in the nucleus. However, the exchange of sera confirmed that SS-A is identical with Ro and SS-B with La [7,8,9].

Since then, countless routine diagnostic immunofluorescent staining patterns of sera of patients with SLE or primary Sjögren’s Syndrome containing autoantibodies to the autoantigen La/SS-B were performed. From these stainings, there is certainly no doubt that La/SS-B is found in nuclei [10]. The nuclear staining pattern of anti-La/SS-B antibodies (Abs) is also in good agreement with the proposed nuclear function(s) of La protein in the context of primary RNA polymerase III transcripts [11,12,13,14,15]. In spite of the nuclear staining pattern of anti-La/SS-B Abs La protein may also have cytoplasmic functions. It was shown that La protein is involved in translational control of 5′ TOP mRNAs and Cap-independent translation of viral and cellular mRNAs harboring internal ribosomal entry sites (IRES elements) (e.g., [16,17,18,19,20,21]).

Looking at the nuclear staining pattern of anti-La Abs leads to the questions of how and when La protein can move to the cytoplasm in order to fulfill these cytoplasmic functions. Already in the eighties, we and many other groups showed a nucleo-cytoplasmic translocation of La protein in cultured cells, which we termed as shuttling of La protein [22,23,24,25,26,27,28,29,30,31,32]. In cell culture, the shuttling of La protein could be induced, for example, by UV irradiation, virus infections, or peroxide exposure. In common, all conditions triggering a shuttling of La protein cause oxidative stress in cultured cells. However, they can also cause cell damage. Moreover, it remains open whether the experimentally used cell culture conditions can be reached in vivo. Therefore, even until today, the shuttling and, thus, a cytoplasmic function of La/SS-B are controversially discussed. Moreover, the driving mechanism for the shuttling of La protein remains unclear.

Recently, we showed that La protein undergoes redox-dependent conformational changes [33]. The primary sequence of La protein contains three cysteine residues being involved in these redox-dependent conformational changes [33,34,35]. So far, oxidoreduction of La protein was experimentally achieved in vitro by exposure of La protein to H_2_O_2_ or CuSO_4_ [33]. Whether or not oxidoreduction and conformational changes also occur in living cells remained unclear. In order to differentiate between the oxidized and reduced forms of La protein, we recently established anti-La monoclonal Abs (anti-La mAbs), which bind to either the reduced or oxidized form of La protein [35]. We found that anti-La mAbs directed to the La motif recognizes the reduced but not the oxidized form of La protein [35]. Vice versa, the epitope recognized by the anti-La mAb 7B6 is cryptic in human La protein but becomes accessible when La protein becomes oxidized. Most importantly, the anti-La mAb 7B6 can be used to detect the oxidized form of La protein by immunofluorescence microscopy (IF) [35]. The anti-La mAb 7B6 recognizes the amino acid sequence EKEALKKIIEDQQESLNK (aa311-328 of human La protein), which includes the α3 helix in the RRM2 domain of La protein [35]. According to previous studies, this region is part of the nuclear retention element of La protein [33,34,35,36,37,38,39].

Here we show that La protein undergoes redox-dependent conformational changes in living cells. After oxidation, La protein shuttles to the cytoplasm. The shuttling of La protein depends on the redox status of the respective cell type. While all analyzed oxidative stress conditions lead to the shuttling of La protein in a wide variety of tested cell types, cultured endothelial cells appeared to be insensitive to the shuttling of La protein. However, deprivation of intracellular reducing agents in endothelial cells turns them sensitive to a redox-dependent shuttling of La protein. Finally, we tested whether or not ligand-/receptor-dependent induced intracellular oxidative stress can also lead to shuttling of La protein. It is well known that toll-like receptor (TLR) signaling results in intracellular oxidative stress. In line with this, we show that translocation of La protein to the cytoplasm can be triggered in a TLR ligand-/receptor-/redox-dependent manner. In summary, we show that La protein is a redox sensor that translocalizes to the cytoplasm in a redox-dependent manner.

## 2. Results

### 2.1. Accessibility of the Epitope Recognized by the Anti-La mAb 7B6 after UV Irradiation

In agreement with recent studies, human cells fixed with methanol lacking peroxides are not stained by the anti-La mAb 7B6 (Figure 1a–c). If the fixed cells, however, are washed with PBS containing H_2_O_2_ (30 μM) the epitope becomes accessible (Figure 1d–f). Already, a short irradiation with UV light prior to the fixation with methanol of analytical grade results in nuclear staining with the anti-La mAb 7B6 without any washing with H_2_O_2_ prior to the staining (Figure 1g–i). These data show that the intracellular oxidative conditions caused by UV irradiation are sufficient to oxidize La protein in living cells whereby the epitope recognized by the anti-La mAb 7B6 becomes accessible. Moreover, in agreement with previous studies, we see that UV irradiation causes a translocation of La protein to the cytoplasm (Figure 1j–l). According to these data, La protein is oxidized when moving to the cytoplasm.

### 2.2. After Oxidation La Protein Remains Intact and Is Not Cleaved during Shuttling to the Cytoplasm

Upstream of the nuclear location signal (NLS) La protein contains potential protease cleavage sites (e.g., [32]). In order to (i) learn whether or not oxidation leads to cleavage of La protein and (ii) confirm the redox-dependent shuttling with an independent approach, two GFP–La fusion proteins were constructed. As schematically shown in Figure 2A, we fused a His-Tag and green fluorescence protein (GFP) either to the N-terminus or C-terminus of human La protein. The two constructs were used for the transfection of human HeLa cells (Figure 2). In case proteolysis of the His-La-GFP fusion protein occurs, we would expect that the cleavage releases a NLS-GFP fragment. As shown previously, such a fusion protein of GFP and the La NLS would be efficiently imported into the nucleus [35]. Vice versa, the GFP–La-His fusion protein should lose the NLS, and therefore, the resulting GFP–La fragment should be localized in the cytoplasm. Consequently, in the case of a proteolytic cleavage as a consequence of oxidation, both the GFP fluorescent staining pattern as well as the protein pattern after SDS-PAGE/immunoblotting should be different for the two GFP–La fusion constructs. As shown in Figure 2, in the absence of H_2_O_2_ both, the His-La-GFP fusion protein (Figure 2B(a–c)) and the GFP–La-His fusion protein (Figure 2B(d–f)) are efficiently imported into the nucleus. Moreover, both constructs show a similar shuttling of the respective GFP–La fusion protein to the cytoplasm, arguing against a different asymmetric cleavage of the two GFP–La fusion constructs under oxidative conditions. This interpretation is supported by the SDS-PAGE/immunoblotting analysis. For this purpose, total extracts of cells transfected with either the His-La-GFP (Figure 2C, left panel) or GFP–La-His (Figure 2C, right panel) were prepared and analyzed by SDS-PAGE/immunoblotting. The cells were either untreated (Figure 2C, -H_2_O_2_) or treated with H_2_O_2_ (Figure 2C, +H_2_O_2_). The extracts were blotted against either the anti-La mAb SW5 (Figure 2C, SW5) or against an anti-GFP Ab (Figure 2C, anti-GFP). The anti-La mAb SW5 recognizes a conformational epitope in the RRM1 of human La protein, which is not sensitive to oxidation [35]. As shown in Figure 2C, there is no evidence for efficient cleavage of either the endogenous La protein or the two different GFP–La fusion proteins. Moreover, there is no evidence for a different asymmetric cleavage of the GFP–La-His or the His-La-GFP fusion protein. The mobility of both the endogenous La protein and the two GFP–La fusion proteins is slightly delayed when treated with H_2_O_2_. The reason for the slight mobility shift is yet unclear. 

As the shift can be seen for endogenous La protein and for both GFP–La fusion proteins this cannot be caused by the fusion partner GFP. We assume that oxidation of La protein somehow influences the 3D structure of La protein and thereby an aberrant mobility and/or thereby causes posttranslational modifications affecting the molecular weight. In contrast to the His-La-GFP fusion protein, the GFP–La-His protein runs as a double band. The reason for this double band remains unclear, but it must be somehow related to the fusion of GFP to the N-terminus of La protein. In summary, we see only some minor degradation of the analyzed GFP–La fusion proteins which, however, cannot explain the cytoplasmic staining of the transfected cells. Consequently, the presented data support the interpretation that La protein is not cleaved under oxidative conditions and cytoplasmic La protein is intact.

### 2.3. Oxidation of La Protein Is the Driving Force for Shuttling of La Protein

Based on the His-La-GFP construct we established mouse 3T3 cells permanently overexpressing human His-La-GFP (Figure 3). Untreated cells were fixed and stained with the anti-La mAb 312B, which recognizes the reduced form of La protein but not the oxidized form [35]. In line with our previous studies the anti-La mAb 312B gives a nuclear staining of untreated His-La-GFP overexpressing mouse 3T3 cells (Figure 3a–c), indicating that La protein is reduced in the nucleus under physiological conditions. In contrast, the anti-La mAb 312B fails to stain fixed untreated His-La-GFP overexpressing mouse 3T3 cells if they are washed with PBS containing H_2_O_2_ prior to the staining (Figure 3d–f, oxidized after fixation). The staining can be restored if the cells are washed with PBS containing ß-mercaptoethanol (Figure 3g–i, reduced after fixation). As soon as living His-La-GFP overexpressing cells were treated with H_2_O_2_ prior to the fixation (Figure 3j–r), His-La-GFP shuttles to the cytoplasm (Figure 3j,m,p). Moreover, the anti-La mAb 312B fails to stain both nuclear and cytoplasmic His-La-GFP (Figure 3k,n). According to these data, His-La-GFP becomes quantitatively oxidized in living cells exposed to H_2_O_2_. The additional oxidation step (by the washing of the fixed cells with PBS containing H_2_O_2_) has no further effect (Figure 3n). Consequently, His-La-GFP and also endogenous La should be completely oxidized if living cells are exposed to H_2_O_2_. As already shown for untreated cells in Figure 3h oxidized La protein in fixed cells can be reduced with PBS containing ß-mercaptoethanol whereby the epitope returns to the configuration recognized by the anti-La mAb 312B. We, therefore, washed His-La-GFP overexpressing cells that were exposed to H_2_O_2_ prior to fixation with PBS containing ß-mercaptoethanol (Figure 3p–r). Indeed, after this treatment, the anti-La mAb 312B is able to stain the His-La-GFP, including the portion of La protein in the cytoplasm (Figure 3q). Complementary data were obtained using the anti-La mAb 7B6 instead of 312B (data not shown). In summary, these data show that La protein shuttles to the cytoplasm as a consequence of a conformational change that is triggered by oxidation.

### 2.4. Further Evidence for Translocation of La Protein to the Cytplasm in Dependence on Oxidative Conditions

As summarized in the introduction section, a variety of conditions had been described which could induce a translocation of La protein to the cytoplasm. So far, we have only provided some qualitative but no quantitative data for the shuttling of La protein in an oxidation-dependent manner. As expected, untreated His-La-GFP overexpressing mouse 3T3 cells (Figure 4A(a–d)) show a nuclear staining both for GFP (Figure 4A(d)) and the anti-La mAb SW5 (Figure 4A(b,c)). In contrast, exposure of the His-La-GFP overexpressing mouse 3T3 cells for 20 min in media supplemented with either H_2_O_2_ or Cu(II)SO_4_ or Fe(II)Cl_2_ or S-Nitrosoglutathione resulted in translocation of La protein to the cytoplasmic compartment (Figure 4A(e–t)). To our knowledge, S-Nitrosoglutathione had not been used so far for triggering a shuttling of La protein from the nucleus to the cytoplasm. We have included S-Nitrosoglutathione in our studies for several reasons. S-Nitrosoglutathione can release nitric oxide (NO). NO is a volatile second messenger involved in the signaling pathway of NFκB, including for example, the signaling pathways of TLRs. NO can oxidize and alkylate SH groups of cysteine residues in proteins. Bearing in mind the sensitivity of the cysteine residue Cys245 in the RRM2 domain of La protein for oxidation and alkylation [40,41,42] we wanted to know whether or not NO can trigger the shuttling of La protein. Indeed, as shown in Figure 4A(q–t), exposing His-GFP–La 3T3 cells to S-Nitrosoglutathione results in a translocation of La protein to the cytoplasm. In summary, all the tested oxidative conditions result in the shuttling of nuclear La protein to the cytoplasm. Next, we estimated the percentage of cells showing a cytoplasmic staining prior to or after exposure to oxidative conditions. According to the quantitative evaluation (Figure 4B), between 30% and 60% of such treated cells show a cytoplasmic staining for La protein.

### 2.5. Nitric Oxide (NO) Mediated Signalling Triggers Shuttling of La Protein

As shown in Figure 4, S-Nitrosoglutathione triggers nucleocytoplasmic shuttling of La protein. Therefore, ligation of TLRs should also result in a shuttling of La protein from the nucleus to the cytoplasm. We challenged this idea by ligation of TLR4 receptors on immune cells with the TLR4 ligand lipopolysaccharide (LPS). For this purpose, we (i) prepared moDCs from monocytes and (ii) isolated a population of monocytes by a single positive selection step using the previously described mAb MDC8. The selected monocyte population expresses the sugar structure 6-sulfo LacNAc (slan epitope), which is recognized by the mAb MDC8 (e.g., [43,44,45,46,47,48,49,50,51]). Slan positive monocytes (slanMo) were selected as they may play an important role as pro-inflammatory cell type in autoimmune diseases, including in lupus erythematosus [44,45].

As shown in Figure 5, untreated moDCs (Figure 5A, untreated) or isolated slanMos (Figure 5B, untreated) show exclusive nuclear staining with the anti-La mAb SW5. However, exposure of moDCs or slanMos to H_2_O_2_, S-Nitrosoglutathione, or LPS results in additional cytoplasmic staining (Figure 5A,B). According to the quantitative evaluation (Figure 5C) around 80% of cells treated with S-Nitrosoglutathione or LPS show a cytoplasmic staining. While TLR4 is located on the cell surface TLR7/8 is an intracellular receptor. Resiquimod^®^ is a ligand of TLR7/8 [49]. Treatment of moDCs or slanMos with Resiquimod^®^ also triggered a shuttling of La protein from the nucleus to the cytoplasm (data not shown).

### 2.6. TLR Receptor Ligand Interactions Are Required for Shuttling of La Protein

Besides wild-type HEK293T cells that are negative for TLR4 (HEK293T-TLR4), a series of genetically manipulated HEK293T cells are commercially available, including HEK293T cells permanently expressing TLR4 (HEK293T+TLR4, see Materials and Methods section). As shown in Figure 6, both fixed HEK293T-TLR4 (Figure 6A, untreated, a–c) and HEK293T+TLR4 cells (Figure 6B, untreated, a–c) give exclusive nuclear staining with the anti-La mAb SW5. In line with the above-described data, both HEK293T-TLR4 (Figure 6A, H_2_O_2_, S-Nitrosoglutathione, d–i) and HEK293T+TLR4 cells (Figure 6B, H_2_O_2_, S-Nitrosoglutathione, d–i) show a shuttling of La protein to the cytoplasm when exposed to H_2_O_2_ or S-Nitrosoglutathione. Moreover, HEK293T+TLR4 cells (Figure 6B, LPS, j–l) also show a ligand-dependent shuttling of La protein to the cytoplasm when exposed to LPS.

In contrast, the HEK293-TLR4 cells (Figure 6A, LPS, j–l) do not respond with a shuttling of La protein to the cytoplasm when exposed to LPS. Neither the treatment of HEK293T-TLR4 cells nor HEK293T+TLR4 cells with the TLR7/8 ligand Resiquimod^®^ triggered a shuttling of La protein (data not shown). Consequently, only in the presence of the TLR4 receptor, the corresponding TLR4 ligand LPS can induce a shuttling of La protein. From these data, we conclude that at least the ligation of the TLR4 ligand LPS to its TLR triggers the shuttling of La protein to the cytoplasm as a consequence of oxidation. According to the quantitative evaluation (Figure 6C), around 80% of the HEK293T cells treated with H_2_O_2_ and between 40 to 60% of the HEK293T cells treated with S-Nitrosoglutathione show a cytoplasmic staining. In contrast, none of the HEK293T-TLR4 cells but around 60% of the HEK293T+TLR4 cells respond with a shuttling of La protein to the cytoplasm when treated with LPS.

### 2.7. The Level of Intracellular Reducing Equivalents Determines the Threshold for Shuttling of La Protein

In agreement with the presented data, fixed untreated human umbilical vein endothelial cells (HUVEC) also produce an exclusive nuclear staining pattern with the anti-La mAb SW5 (Figure 7A(a–c)). Unexpectedly, however, neither exposure to H_2_O_2_ nor Cu(II)SO_4_ nor Fe(II)Cl_2_ nor S-Nitrosoglutathione nor LPS resulted in cytoplasmic staining of the HUVECs with the anti-La mAb SW5. Although exposed to these various oxidative conditions, all treated cells showed an exclusive nuclear staining pattern (Figure 7A(d–r)). We speculated that the intracellular level of reducing equivalents such as glutathione might be higher in the cultured HUVECs protecting La protein against oxidation. To challenge this assumption, HUVECs were grown in the presence of buthionine-sulfoximine (BSO), which inhibits the intracellular synthesis of glutathione and can lead to a reduction of intracellular glutathione of more than 90% [52,53]. As expected, untreated HUVECs grown under these conditions still give an exclusive nuclear staining pattern with the anti-La mAb SW5 (Figure 7B(a–c)). However, now exposure to H_2_O_2_ or Cu(II)SO_4_ or Fe(II)Cl_2_ or S-Nitrosoglutathione or LPS leads to cytoplasmic staining of the HUVECs with the anti-La mAb SW5 (Figure 7B(d–r)). The quantitative evaluation confirms the IF data (Figure 7C). From these data, we conclude that the intracellular level of reducing equivalents buffer oxidizing reagents and thereby determines the threshold up to which La protein is protected against oxidation and stays in the nucleus. Consequently, La protein is a redox sensor, which shuttles to the cytoplasm when the intracellular reducing equivalents are consumed, and remaining oxidative reagents can oxidize La protein.

## 3. Discussion

The autoantigen La/SS-B is a nuclear protein in interphase cells (e.g., [8,9,10,33,34,35,54]). According to numerous publications, however, La protein is also involved in cytoplasmic processes such as translational control of viral and cellular mRNAs [16,17,18,19,55,56,57,58,59,60,61,62,63,64,65,66,67,68,69,70,71,72] leading to the question: what is the triggering mechanism for translocation of La protein from the nucleus into the cytoplasm? Already in the 1980s, we and many others observed translocation of La protein to the cytoplasm, which we termed as the shuttling of La protein (e.g., [22,23,24,25,26]). Unfortunately, the shuttling was not seen with all anti-La Abs and all types of cells tested and thus could not be confirmed in all laboratories. Therefore, the shuttling of La protein was controversially discussed, including as a leakage of La protein, e.g., during the preparation of nuclei or fixation of cells [11]. Similarly, the association of La protein with mRNAs was in part interpreted as an artificial interaction occurring during the preparation of total cellular extracts and/or co-immunoprecipitation.

In addition, structural analyses did not support the proposed function of La protein in IRES-dependent translation [36,37,38]. For example, Craig et al. provided in vitro experimental evidence that IRES-dependent translation of poliovirus mRNAs requires dimerization of La protein [55]. However, structural analyses did not support a dimerization of La protein [36,37,38]. Noteworthy to mention is that even today, the structure of full-length La protein is not completely known. So far, only three domains of La protein were published, including the structure of the La motif, the RRM1, and the RRM2 domain of La protein [36,37,38]. All of these structural data were collected under reducing conditions. Moreover, the ultracentrifugation step from which it was concluded that La protein is not able to form dimers was performed in the presence of reducing agents.

In independent reports, however, it was shown that a fragment of La protein containing the two cysteine residues Cys232 and Cys245 in the RRM2 domain is highly sensitive to oxidation and alkylation [40,41,42]. Based on this La fragment, Huth et al. even developed an assay that was termed as ALARM NMR [40,41,42]. The abbreviation ALARM NMR stands for a La assay to detect reactive molecules by nuclear magnetic resonance. This assay was established to identify oxidative and alkylating capabilities of novel drugs. Recent data from our lab show that not only this La protein-derived fragment used for the ALARM NMR assay but the whole La protein is sensitive to oxidation. Besides the two cysteine residues, Cys232 and Cys245, in the RRM2, the cysteine residue Cys18 in the La motif contributes to the redox sensitivity of La protein. According to these data, La protein undergoes reversible redox-dependent conformational changes, including the formation of dimers.

The redox status of La protein can be followed using mAbs, specifically recognizing either the reduced or the oxidized form of La protein. The four anti-La mAbs 312B, 2F9, 32A, and 27E react with the reduced form of La protein but fail to react with the oxidized form of La protein. All these anti-La mAbs recognize a conformational epitope in the La motif. The redox sensitivity of these anti-La mAbs is dependent on the cysteine residue Cys18. Vice versa, the epitope recognized by the anti-La mAb 7B6 is cryptic in the reduced La form but becomes accessible under oxidative conditions. The epitope recognized by the anti-La mAb 7B6 includes the α3 helix in the RRM2 domain, which is part of the previously described nuclear retention element [73]. We have used the 7B6 epitope as a protein tag in a series of studies [28,74,75,76,77,78,79,80,81,82,83,84,85,86,87,88,89,90,91,92,93]. According to these data, the 7B6 epitope itself is not sensitive to oxidation. 

Our current explanation for the redox-dependent accessibility of the 7B6 epitope is as follows: Alkylation of the cysteine residue Cys232 or the formation of a disulfide bridge with the cysteine residue Cys245 (as it occurs in the ALARM-NMR assay) or the formation of a disulfide bridge with the cysteine residue Cys18 in the same La protein or an independent La protein molecule alters the structure or the accessibility of the α3 helix in the RRM2 domain of La protein for the anti-La mAb 7B6. Accordingly, oxidative conditions lead to allosteric conformational changes in the La protein that interfere with the function of the nuclear retention element and, as a consequence, could consequently trigger the translocation to the cytoplasm.

In the current manuscript, we therefore wanted to verify if these redox-dependent changes and the shuttling of La protein also occur in living cells. Moreover, we wanted to learn if the shuttling of La protein is a more common response to oxidative stress that occurs in all kind of cells or is limited to certain cell types. This question came up from the controversial data reported in the literature about the shuttling of La protein. These discrepancies could be explained as the result of the analyzed different cell types or simply be caused by the growing conditions, e.g., by the concentration of reducing agents in the different cell culture media.

In agreement with previous studies, cells cultured in the absence of oxidative stress resulted in nuclear staining with the anti-La mAb 312B, while the anti-La mAb 7B6 fails to stain, indicating that La protein exists in its reduced form under these conditions and is mainly located in the nucleus. However, when living cells were exposed to oxidative stress, the anti-La mAb 7B6 was able to stain them while now the anti-La mAb 312B did not detect the reduced form of La protein. Vice versa, when the fixed cells were washed under reducing conditions, La protein could be detected with the anti-La mAb 312B but then the anti-La mAb 7B6 failed to stain the cells. Therefore, we concluded that under physiological growing conditions, La protein is nuclear and exists in a reduced form but oxidative stress conditions lead to conformational changes.

To independently confirm the presence and localization of La protein cells were stained with the anti-La mAb SW5, which recognizes a conformational epitope in the RRM1 domain that is not sensitive to oxidation. In addition, we confirmed the staining pattern of the anti-La mAbs by double immunofluorescence microscopy with the staining of GFP–La fusion proteins. To rule out that the fusion of GFP to La protein affects the localization and shuttling of La protein, we fused GFP to either the N-terminus or C-terminus of La protein. The fusion of GFP to the N- or C-terminus of La protein does not affect the redox-dependent staining pattern of the respective GFP–La fusion protein. Therefore, the two GFP–La protein fusion variants cannot be asymmetrically cleaved in a redox-dependent manner. This tells us that La protein does not lose the C-terminally located NLS during shuttling due to proteolysis and, thus, the cytoplasmic La protein is intact, full-length oxidized La protein. This interpretation is supported by the results from SDS-PAGE/immunoblotting.

However, this data has one disadvantage. It remains unknown whether the intracellular conditions triggering the shuttling of La protein in cultured cells can also occur in vivo. For example, can the concentration of hydrogen peroxide or heavy metal ions occur in vivo? We therefore tested NO as an additional oxidative agent. NO became of interest to us for several reasons: (i) NO is able to alkylate sensitive cysteine residues including in proteins, and La protein contains cysteine residues highly sensitive to alkylation and oxidation [40,41,42]. (ii) Moreover, NO is used in the NFκB signaling pathway as a volatile second messenger of many immune receptors of both the innate and adaptive immune response. Representatively, we looked at the TLR signaling pathway. Thereby we showed that TLR4 ligand–receptor interactions can indeed lead to the shuttling of La protein. The TLR-dependent signaling is specific as cells lacking the TLR4 receptor did not respond to exposure of the TLR ligand although all oxidative conditions, including NO, lead to the shuttling of La protein.

In summary, we were able to show that the shuttling of La protein can be induced in all analyzed cell types under all the various oxidative stress conditions tested. Therefore, it should be a common oxidative stress response in all cell types. The shuttling of La protein was verified with three different anti-La mAbs recognizing three different domains of La protein, including complementary redox-dependent epitopes and redox-independent epitopes. In addition, the shuttling was confirmed by double-labeling the anti-La mAbs with GFP–La fusion proteins. Moreover, we showed that the shuttling of La protein depends on the amount of intracellularly available reducing equivalents. We found that the shuttling is triggered when the intracellular reducing equivalents are depleted. Thus, the growing conditions and the intracellular content of reducing agents directly influence the threshold for the La shuttling, which may also be the reason for the reported controversial data with respect to the shuttling of La protein. In this context, it is also important to mention that antioxidant protection is impaired in SLE patients [94,95,96,97,98,99]. Consequently, the threshold for oxidative stress and thus for the translocation of La protein to the cytoplasm should be lower in SLE patients.

Based on our data, we hypothesize that after consumption of intracellular reducing equivalents, La protein will be oxidized in the nucleus, which then results in its translocation to the cytoplasm. The functional reason might be: In the cytoplasm, oxidized dimers of La protein interact with IRES elements of certain mRNAs, facilitating their translation. Following this interpretation, La protein is a redox sensor involved in the expression of certain mRNAs, which respond to oxidative stress after the consumption of intracellular reducing agents. In agreement with this hypothesis, the reduced La protein is almost exclusively found in the nucleus in the absence of oxidative stress. Soon after the occurrence of oxidative stress, oxidized La protein can be detected in both compartments, the nucleus, and the cytoplasm. 

We, therefore, assume that La protein is oxidized in both compartments including in the nucleus. The oxidized La protein can leave the nucleus because the conformational changes caused by oxidation interferes with the interaction of La protein with a nuclear binding partner via its nuclear retention element, as indicated by the availability of the 7B6 epitope. As we do not see a rapid degradation of La protein under oxidative stress conditions, we expect that oxidized La protein can also be reduced after oxidative stress and re-enter into the nucleus. From our current work, however, we cannot conclude if oxidized La protein has to be reduced prior to the import into the nucleus.

One of our most interesting novel findings is that the shuttling of La protein can also be triggered via ligation of immune receptors and as a result of NO-mediated oxidation. Bearing in mind that the intracellular reducing equivalents are impaired in cells of patients suffering from rheumatic diseases [94,95,96,97,98,99], we present here for the first time a link between the autoantigen La/SS-B and an altered key signaling pathway of both the innate and adaptive immune system in autoimmune patients.

## 4. Materials and Methods

### 4.1. Cell Lines and Cell Culture

If not noted otherwise, cell lines were cultured in Dulbecco’s Modified Eagle Medium (DMEM, Gibco, Thermo Fisher Scientific, Germany), supplemented with 10% (*v*/*v*) FBS (Sigma-Aldrich, Darmstadt, Germany, 2 mM L-glutamine, 100 U/mL of penicillin, and 100 μg/mL of streptomycin (Biochrom, Berlin, Germany) and maintained at 37 °C in a humidified atmosphere of 5% CO_2_. The HEK293 cell line (293-htlr4md2cd14) stably transfected with human TLR4a, MD2 and CD14 were purchased from InvivoGen (München, Germany). Cells were cultured according to the manufacturer´s instructions. Human Umbilical Vein Endothelial Cells (HUVEC, Catalog Number C-003-5C) were purchased from Invitrogen (Thermo Fisher Scientific Life Technologies GmbH, Darmstadt, Germany). Cell culture was performed according to the manufacturer. In order to inhibit the synthesis of glutathione, the endothelial cells were treated with buthionine-sulfoximine (BSO, Sigma-Aldrich Chemie GmbH, Steinheim, Germany) for 24 h at a concentration of 100 μM. HeLa cells expressing fusion constructs consisting of green fluorescence protein (GFP) and human La protein were prepared by transfection. For this purpose, two GFP fusion constructs were cloned: In the one construct, we fused the His-Tag to the N-terminus of La protein and the GFP reading frame to the C-terminus of La protein (His-La-GFP). In the other construct, we fused the GFP reading frame to the N-terminus of human La protein and the His-tag to the C-terminus of La protein (GFP–La-His) (e.g., [35]). The La protein or GFP reading frame was amplified by PCR and cloned into either pEGFP-C2 or the previously described La-His construct in the vector pCI [100,101]. Buffy coats of anonymous healthy blood donors were provided by the German Red Cross (Dresden, Germany) after the written consent of the donors. 

All procedures using human materials were performed in accordance with local regulations, guidelines and approval from the local ethics committee of the Medical Faculty Carl Gustav Carus Dresden of the TU Dresden (EK27022006). Peripheral blood mononuclear cells (PBMCs) were isolated from buffy coats as described by Feldmann et al. [102]. Monocyte-derived dendritic cells were prepared from monocytes that were isolated from PBMCs via a positive selection using CD11 beads as described by Hauptmann et al. [103]. 6-sulfo-LacNAc^+^ monocytes (slanMos) were isolated from PBMCs using the mAb MDC8 as described by Wehner et al. [50]. For immunofluorescence microscopical studies, the cells were either cultured in eight-well chamber slides or 24 well plates containing a sterilized coverslip. For chamber slides, the adherent cells were seeded at a cell density of 0.5 × 10^5^ in 200 µL of DMEM media per chamber. MoDCs and slanMos were seeded at a cell density of 0.5 × 10^6^ in 200 µL of RPMI media per chamber. For UV irradiation experiments 10^5^ cells were seeded per well of a 24 well plate. Cells were allowed to grow for 24 h at 37°C in a humidified atmosphere of 5% CO_2_.

### 4.2. Induction of Shuttling

In order to trigger the shuttling of La protein, cells grown in chamber slides were exposed to the respective oxidative reagent for 20 min in 100 µL of PBS. Thereafter, the cells were fixed. If not noted otherwise, the used concentration of the respective triggering reagent was: H_2_O_2_ (160 mM), Cu(II)SO_4_ (1 mM), Fe(II)Cl_2_ (1 mM), S-Nitrosoglutathione (1 mM) (N4148) and LPS (0.05 mM) (O111:B4) (Sigma-Aldrich Chemie GmbH, Steinheim, Germany), and Resiquimod^®^ (0.2 mM) (ENZO^®^ Life Science GmbH, Lörrach, Germany).

For UV irradiation, cells grown on a coverslip in a 24-well plate were used. UV irradiation was performed using an UVC 500 Ultraviolet Crosslinker (Hoefer; GE Healthcare, Freiburg, Germany). Prior to irradiation, cell culture media was replaced by 200 µL PBS. The lid of the 24-well plate was removed, and the cells were irradiated with 250 mJ/m^2^. After UV exposure, PBS was replaced by media, and the cells were kept at 37 °C and 5% CO_2_. The cells were fixed 9 min or 2 h after UV irradiation.

### 4.3. Fixation and Staining of Cells for Immunofluorescence Microscopy

To avoid oxidation of La protein during the fixation procedure, the cells were fixed with methanol of analytical grade delivered and stored in a brown glass bottle (Merck KGaA, Darmstadt, Germany) [33]. Cells were fixed for at least 1 h or overnight. Prior to the staining, the cells were rehydrated with PBS (5 min) and stained with the respective anti-La mAb (e.g., [34,35,104,105]). For the oxidation of fixed cells, cells were washed with PBS containing H_2_O_2_ (160 mM). For reduction of fixed cells, cells were washed with PBS containing ß-mercaptoethanol (5 mM). Cells were documented using an Axiovert M200 epifluorescence microscope and the software AxioVision (Carl Zeiss Microscopy GmbH, Jena, Germany). For quantitative analysis, we used a Keyence microscope and the software package BZ Analyzer II (Keyence Microscope Europe, München, Germany).

### 4.4. Anti-La mAbs

In the present study, we used cell culture supernatants of hybridomas secreting the previously described anti-La mAbs SW5 [104,105], 7B6 [34,35], and 312B [35] for immunofluorescence (IF) microscopy, and SDS-PAGE/immunoblotting.

### 4.5. SDS-PAGE/Immunoblotting

Total extracts were prepared and analyzed by SDS-PAGE/immunoblotting as described previously (e.g., [106,107]).

### 4.6. Statistical Analysis

In general, if not noted otherwise, all experiments were performed at least in triplicate. For quantitative evaluation, images of five independent fields were taken and evaluated. Values are expressed as mean ± SEM. For statistical evaluation, one-way ANOVA followed by Holm–Sidak multiple comparisons test was performed using GraphPad Prism version 7.00 for Windows, (GraphPad Software, La Jolla, CA, USA). *p* Value < 0.0001 = ****.

## Figures and Tables

**Figure 1 ijms-22-09699-f001:**
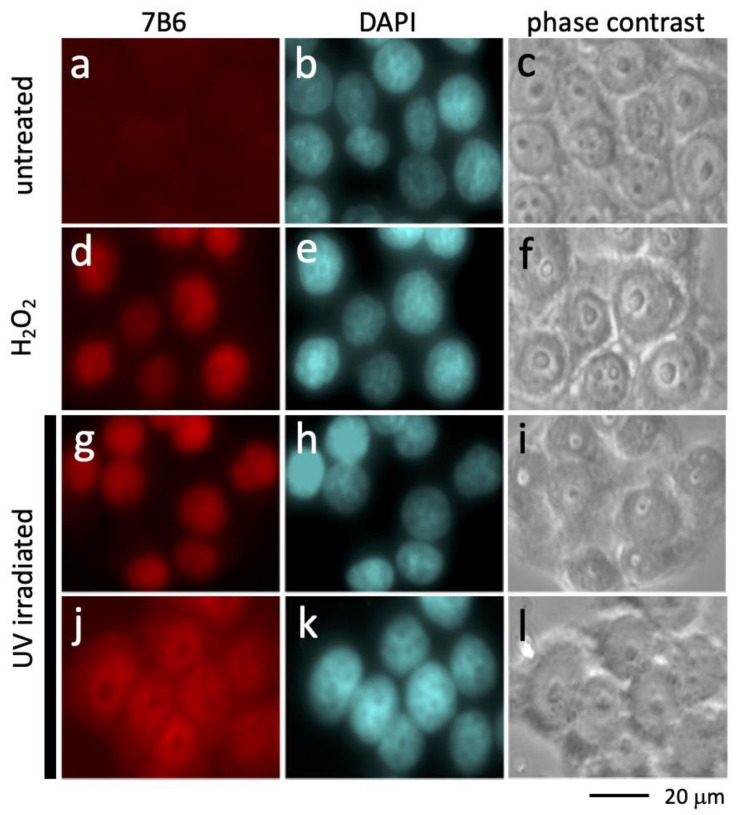
Translocation of La protein to the cytoplasm after UV irradiation. Fixed cells were stained with (**a**) the anti-La mAb 7B6 or (**b**) DAPI. (**c**) Phase-contrast image. (**d**) Prior to the staining with the anti-La mAb 7B6 La protein was oxidized by washing the fixed cells with PBS containing H_2_O_2_. (**e**) DAPI staining of fixed cells. (**f**) Phase-contrast image. (**g**) Cells were UV irradiated. Nine min after irradiation, the cells were fixed and stained with the anti-La mAb 7B6. (**h**) DAPI staining of fixed cells. (**i**) phase-contrast image. (**j**) Cells were UV irradiated. Two h after irradiation, the cells were fixed and stained with the anti-La mAb 7B6. (**k**) DAPI staining of fixed cells. (**l**) Phase contrast image.

**Figure 2 ijms-22-09699-f002:**
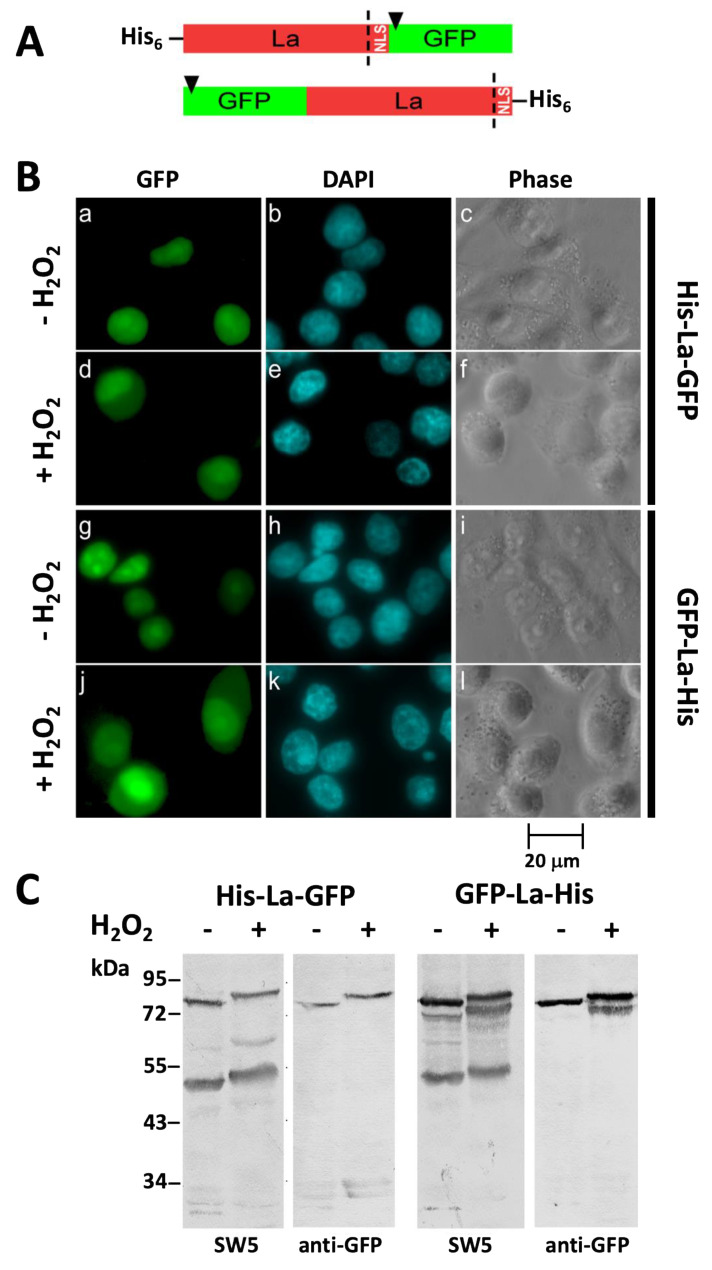
La protein is not cleaved under oxidative conditions. (**A**) La protein contains potential proteolytic cleavage sites (dashed line) upstream of the nuclear location signal (NLS). Two different GFP–La-His fusion proteins were constructed in which GFP is fused either upstream or downstream of the La reading frame. In case GFP is fused to the N-terminus of La protein, a His-Tag is fused to the C-terminus of La protein resulting in the fusion protein GFP–La-His. In case GFP is fused to the C-terminus of La protein a His-Tag is fused to the N-terminus of La protein resulting in the fusion protein His-La-GFP. (**B**) Human HeLa cells expressing either the His-La-GFP fusion construct (**a**–**f**) or the GFP–La-His fusion construct (**g**–**l**). Cells were either untreated (**a**–**c**; **g**–**i**) or treated with H_2_O_2_ (**d**–**f**; **j**–**l**). (**C**) SDS-PAGE and immunoblotting of total extracts from cells expressing either the His-La-GFP fusion construct (**His-La-GFP**) or the GFP–La-His fusion construct (**GFP-La-His**). Total extracts were blotted against either the anti-La mAb SW5 (**SW5**) or an anti-GFP Ab (**anti-GFP**). Extracts were obtained from transfected cells which were either untreated (−) or treated with H_2_O_2_ (+) prior to the preparation of the extracts.

**Figure 3 ijms-22-09699-f003:**
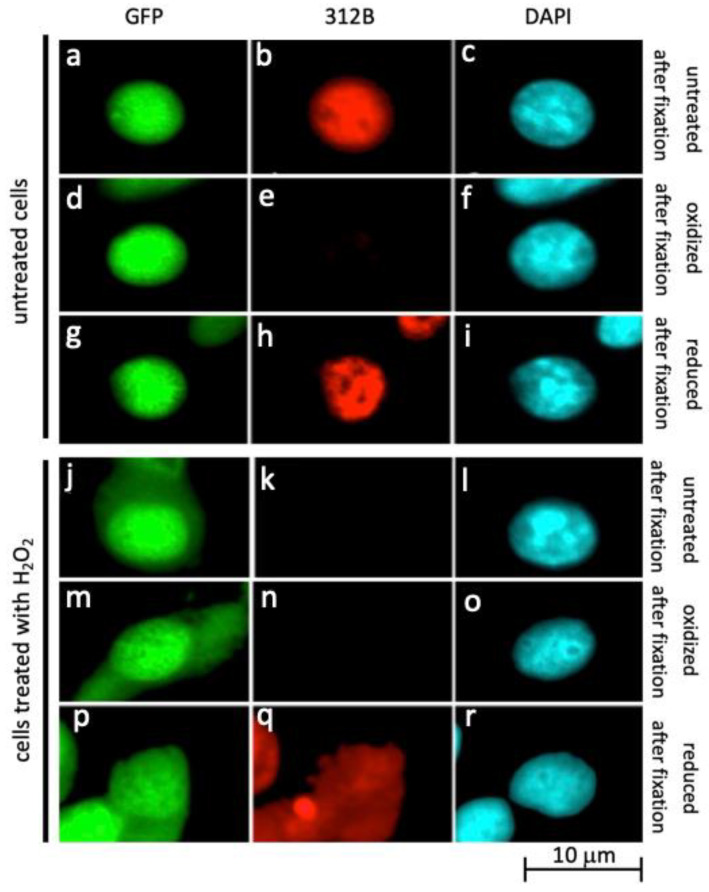
La protein is oxidized in the cytoplasm. (**a**–**q**) IF microscopy of mouse 3T3 cells overexpressing His-La-GFP (**a**,**d**,**g**,**j**,**m**,**p**) stained with the anti-La mAb 312B (**b**,**e**,**h**,**k**,**n**,**q**) or DAPI (**c**,**f**,**i**,**l**,**o**,**r**). (**a**–**c**) Untreated fixed cells. (**d**–**f**) Untreated cells oxidized with H_2_O_2_ after fixation. (**g**–**i**) Untreated cells oxidized with H_2_O_2_ after fixation followed by a reduction with ß-mercaptoethanol. (**j**–**l**) Oxidized living cells, untreated after fixation. (**m**–**o**) Oxidized living cells, oxidized after fixation. (**p**–**r**) Oxidized living cells, reduced after fixation.

**Figure 4 ijms-22-09699-f004:**
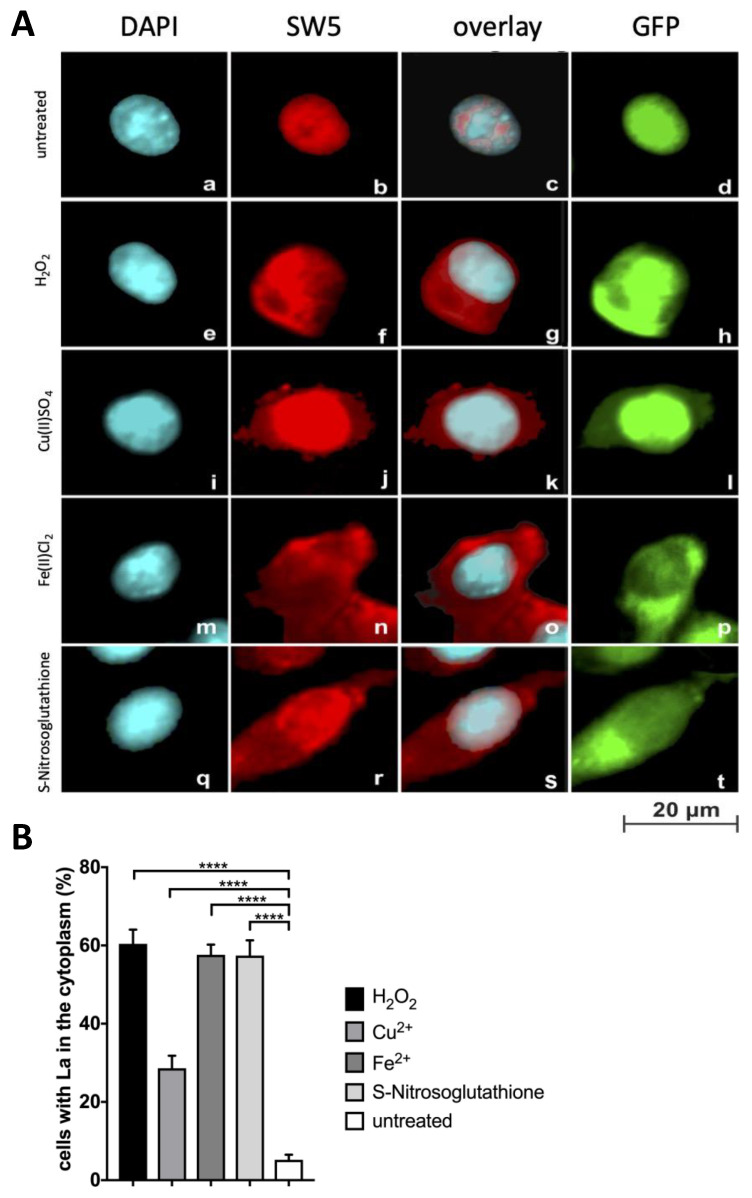
Translocation of La protein to the cytoplasm and its dependence on oxidative conditions. (A) Immunofluorescence analysis. (**a**–**d**). Untreated mouse 3T3 cells overexpressing His-La-GFP. (**e**–**h**) Cells treated with H_2_O_2_ or (**i**–**l**) CuSO_4_, or (**m**–**p**) FeCl_2,_ or (**q**–**t**) S-Nitrosoglutathione. Fixed cells were stained with (**a**,**e**,**i**,**m**,**q**) DAPI or (**b**,**f**,**j**,**n**,**r**) the anti-La mAb SW5. Overlay of DAPI and SW5 staining (**c**,**g**,**k**,**o**,**s**) as well as GFP fluorescence (**d**,**h**,**l**,**p**,**t**). (**B**) Quantitative evaluation of the percentage of cells showing cytoplasmic staining comparing untreated cells with cells after the respective treatment. *p* Value < 0.0001 = ****.

**Figure 5 ijms-22-09699-f005:**
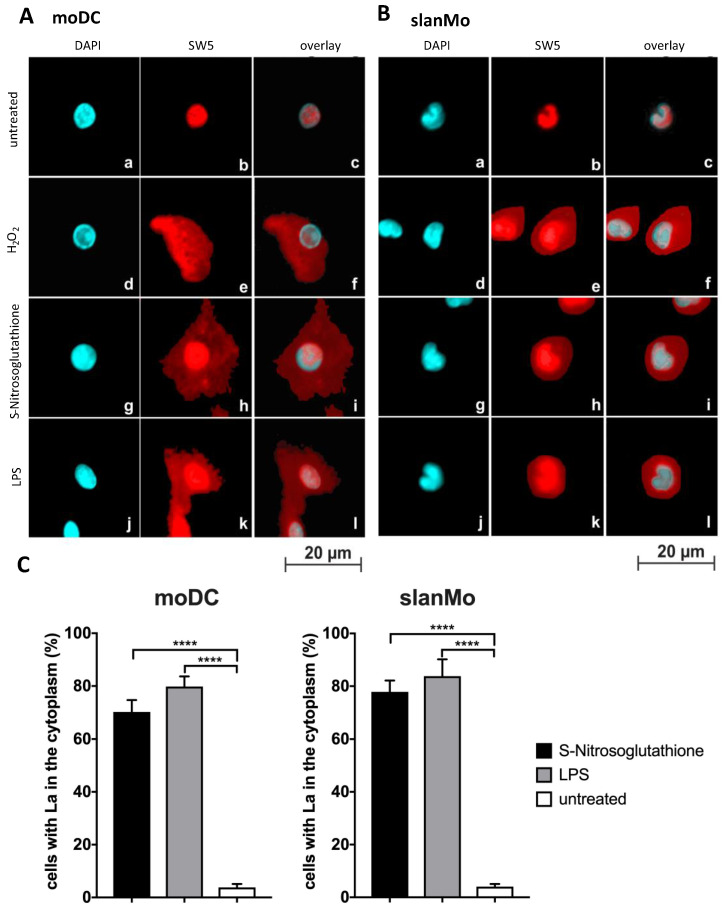
Shuttling of La protein in moDCs or slan monocytes (slanMos). (**A**,**B**) MoDCs and slanMos were stained with DAPI (**a**,**d**,**g**,**j**) or the anti-La mAb SW5 (**b**,**e**,**h**,**k**). The overlay of both channels is shown (**c**,**f**,**i**,**l**). (**a**–**c**) Untreated cells. (**d**–**f**) Cells treated with H_2_O_2_. (**g**–**i**) Cells treated with S-Nitrosoglutathione. (**j**–**l**) Cells treated with LPS. (**C**) Quantitative evaluation. *p* Value < 0.0001 = ****.

**Figure 6 ijms-22-09699-f006:**
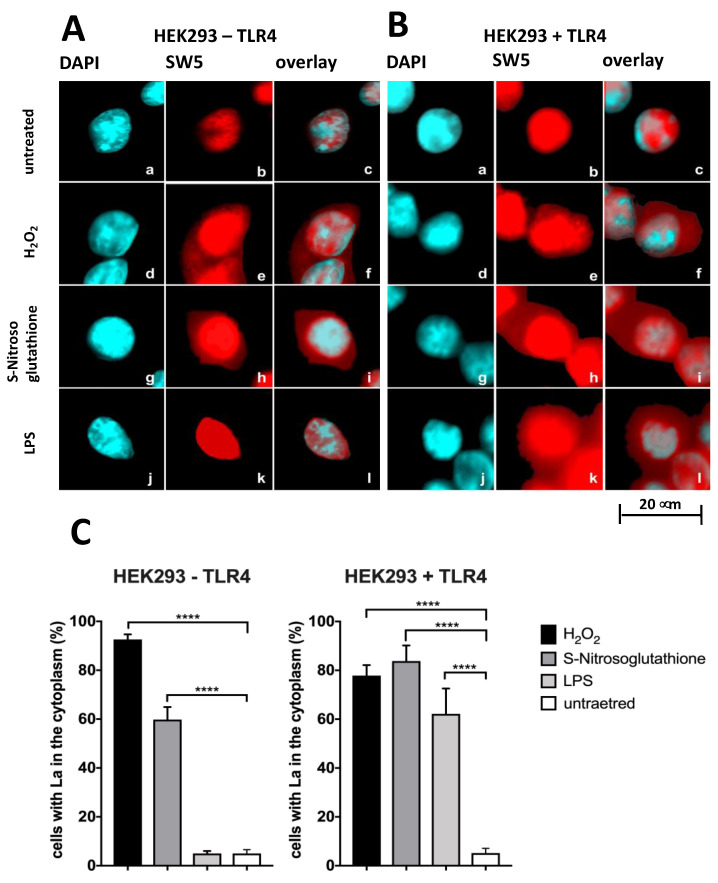
Shuttling of La protein in dependence on Toll-like Receptor/Ligand ligation**.** (**A,B**) HEK293T cells lacking (HEK293T-TLR4) or overexpressing (HEK293T+TLR4) TLR4 were stained with DAPI (**a**,**d**,**g**,**j**) or the anti-La mAb SW5 (**b**,**e**,**h**,**k**). Overlay of both is shown (**c**,**f**,**i**,**l**). (**a**–**c**) Untreated cells. (**d**–**f**) Cells treated with H_2_O_2_. (**g**–**i**) Cells treated with S-Nitrosoglutathione. (**j**–**l**) Cells treated with LPS. (**C**) Quantitative evaluation. *p* Value < 0.0001 = ****.

**Figure 7 ijms-22-09699-f007:**
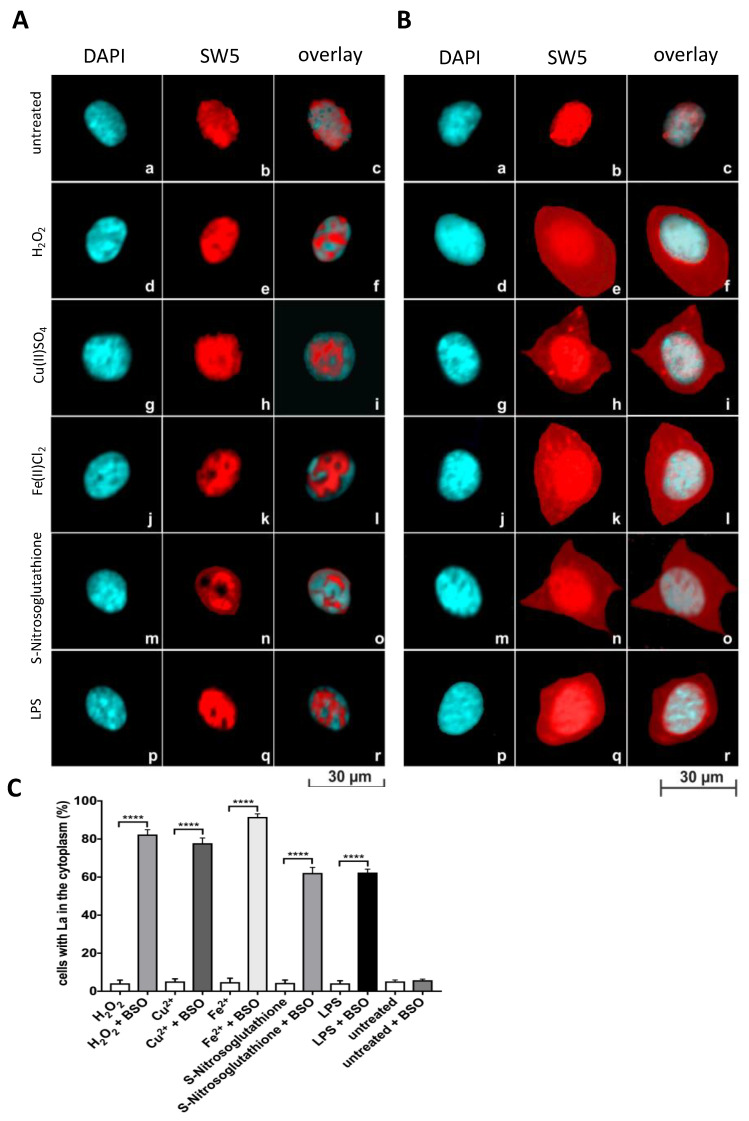
Shuttling of La protein in human umbilical vein endothelial cells (HUVEC). (**A,B**) HUVECs grown in the absence (**A**) or presence (**B**) of buthionine-sulfoximine (BSO). HUVECs were stained with DAPI (**a**,**d**,**g**,**j**,**m**,**p**) or the anti-La mAb SW5 (**b**,**e**,**h**,**k**,**n**,**q**). Overlay of both channels is shown in addition (**c**,**f**,**i**,**l**,**o**,**r**). (**a**–**c**) Untreated cells. (**d**–**f**) Cells treated with H_2_O_2_. (**g**–**i**) Cells treated with Cu(II)SO_4_. (**j**–**l**) Cells treated with Fe(II)Cl_2_. (**m**–**o**) Cells treated with S-Nitrosoglutathione. (**j**–**l**) Cells treated with LPS. (**C**) Quantitative evaluation. *p* Value < 0.0001 = ****.

## Data Availability

Original data are available on request from M.P.B.

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
