# Peer review of "And Yet It Moves: Oxidation of the Nuclear Autoantigen La/SS-B Is the Driving Force for Nucleo-Cytoplasmic Shuttling"

_ijms, 2021, doi:10.3390/ijms22189699_

Round 1
Reviewer 1 Report
In a word, since the Reichlin/Tan/Provost days, anti-La is often encountered, but knowledge of what this molecule is, how it is controlled, and what is its role as an effector or respondent in abnormal human biology.
In this sense, this is first extensive attempt to answer the questions. It is authored by very trustworthy investigators. Assuming that the science is correct, this paper may shift the paradigm of the meaning of autoantibodies in human pathology.
Author Response
Reviewer 1:
Yes Can be improved Must be improved Not applicable
Does the introduction provide sufficient background and include all relevant references?
(x) ( ) ( ) ( )
Is the research design appropriate?
(x) ( ) ( ) ( )
Are the methods adequately described?
(x) ( ) ( ) ( )
Are the results clearly presented?
(x) ( ) ( ) ( )
Are the conclusions supported by the results?
(x) ( ) ( ) ( )
Comments and Suggestions for Authors
In a word, since the Reichlin/Tan/Provost days, anti-La is often encountered, but knowledge of what this molecule is, how it is controlled, and what is its role as an effector or respondent in abnormal human biology.
In this sense, this is first extensive attempt to answer the questions. It is authored by very trustworthy investigators. Assuming that the science is correct, this paper may shift the paradigm of the meaning of autoantibodies in human pathology.
Response to reviewer 1:
We can only be very grateful to such an optimal evaluation of our ms.

Reviewer 2 Report
It is a well-done paper in which the authors study the driving mechanism for the shuttling of La protein. They use different cells to obtain the data, so they assume that all the cells have the same performance under the same experimental conditions. Please, clarify or discuss this issue.
The authors demonstrate the shuttling of La protein depends on the redox status of the respective cell type. There is a translocation of La protein from the nucleus to the cytoplasm. It can be assumed that also a reverse process can be held?
Author Response
Response to Reviewers:
Reviewer 2:
Yes Can be improved Must be improved Not applicable
Does the introduction provide sufficient background and include all relevant references?
(x) ( ) ( ) ( )
Is the research design appropriate?
(x) ( ) ( ) ( )
Are the methods adequately described?
(x) ( ) ( ) ( )
Are the results clearly presented?
(x) ( ) ( ) ( )
Are the conclusions supported by the results?
(x) ( ) ( ) ( )
Comments and Suggestions for Authors
Reviewer 2:
It is a well-done paper in which the authors study the driving mechanism for the shuttling of La protein.
Response to Reviewer 2:
As for reviewer 1, again we are very grateful for the overall very positive evaluation of our ms.
Reviewer 2:
They use different cells to obtain the data, so they assume that all the cells have the same performance under the same experimental conditions. Please, clarify or discuss this issue.
Response Reviewer 2:
In the literature, the shuttling of La protein is controversially discussed. There are two possible reasons for these discrepancies: (i) The different groups used different cell types and the different cell types respond differently, and (ii) there is the chance that simply the growing conditions, meaning the media used, differed especially with respect to the content of reducing agents. To rule out the problem of cell type, we tried to induce the shuttling in a variety of cell lines. According to our data, we could induce a shuttling of La protein by a variety of oxidative stress conditions in all cell lines tested. Therefore, the shuttling of La is not an artefact but a common response to oxidative stress that can occur in all cell types. According to our results, the amount of intracellular reducing agents, however, can effect the threshold for the shuttling including an inhibition of the shuttling. Consequently, the controversial data are most likely due to the different growing conditions (and also the use of non-redox dependent anti-La abs).
In order to discuss this minor comment we have modified the section (see lanes 440 to 464 on pages 16 and 17) of the discussion related to this point as follows:
In summary, we were able to show that the shuttling of La protein can be induced in all the analyzed different cell types under all the various oxidative stress conditions tested. Therefore, it should be a common oxidative stress response in all cell types. The shuttling of La protein was verified with three different anti-La mAbs recognizing three different domains of La protein including complementary redox-dependent epitopes and redox-independent epitopes. In addition, the shuttling was confirmed by double labeling of anti-La mAbs with GFP-La fusion proteins. Moreover, we showed that the shuttling of La protein depends on the amount of intracellularly available reducing equivalents. We found that the shuttling is triggered when the intracellular reducing equivalents are depleted. Thus, the growing conditions and the intracellular content of reducing agents directly influences the threshold for the La shuttling which may also be the reason for the reported controversial data with respect to the shuttling of La protein. In this context it is also important to mention that antioxidant protection is impaired in SLE patients [94-99]. Consequently, the threshold for oxidative stress and thus for translocation of La protein to the cytoplasm should be lower in SLE patients.
Reviewer 2:
The authors demonstrate the shuttling of La protein depends on the redox status of the respective cell type. There is a translocation of La protein from the nucleus to the cytoplasm. It can be assumed that also a reverse process can be held?
Response to Reviewer 2:
This is a very interesting question. Indeed we assume that La protein can also return into the nucleus. From the fact that in the absence of oxidative stress (reduced) La protein enters in the nucleus where it is retained until oxidation we expect that reduction of La occurs prior or after entering into the nucleus. However, we cannot finally rule out that oxidized La protein has to be reduced prior to the import into the nucleus. At least it appears unlikely that oxidized La is degraded as we don’t see an increase of degradation products under oxidative conditions.
In order to discuss also this minor comment we have modified the section (see lanes 471 to 481 on page 17) of the discussion related to this point as follows:
In agreement with this hypothesis reduced La protein is almost exclusively found in the nucleus in the absence of oxidative stress. Soon after occurrence of oxidative stress oxidized La protein can be detected in both compartments the nucleus and the cytoplasm. We therefore assume that La protein is oxidized in both compartments including in the nucleus. The oxidized La protein can leave the nucleus because the conformational changes caused by oxidation interferes with the interaction of La protein with a nuclear binding partner via its nuclear retention element as indicated by the availability of the 7B6 epitope. As we do not see a rapid degradation of La protein under oxidative stress conditions we expect that oxidized La protein can also be reduced after oxidative stress and re-enter into the nucleus. From our current work, however, we cannot conclude if oxidized La protein has to be reduced prior to the import into the nucleus.

This manuscript is a resubmission of an earlier submission. The following is a list of the peer review reports and author responses from that submission.
Round 1
Reviewer 1 Report
This is a sound study providing new insight into the role of La protein as a redox sensor, which shuttles to the cytoplasm after conformational changes triggered by its oxidation due to a variety of stimuli. These include immunological stimuli such as TLR4 ligand receptor interactions, providing new link between La/SS-B autoantigen and altered immunological signaling pathways. Overall I think Authors' conclusions are supported by experimental data and discussed in a balanced way.
I only suggest a minor correction: I think that on line 502 "...and an altered impaired key signaling pathway..." either altered or impaired is redundant and should be deleted.
Author Response
Response to reviewer:
We are grateful for this kind review report. As requested we have deleted the term “impaired” in line 502 of the ms.
Reviewer 2 Report
This is a comprehensive, if not exhaustive, demonstration that oxidative stress drives La/SSB nuclear/cytoplasmic translocation. For the most part, the methods are generally sound. This represents an incremental advance from previous work.
Criticisms:
1) The presentation of data is entirely and rather cursorily descriptive; more like a narrative than the typical scientific report. There are no indications of how many times the experiments were repeated ("n'), data distributions (error bars) nor any attempt at analyzing intergroup differences. The absence of knowing what the data really shows (n, and dispersion) limits the credibility of these findings.
2) Many methods were either inadequately detailed and/or primarily presented in figure legends, and sometimes results, rather than in Methods section per se. In particular, the sources of oxidants, TLR ligands, and methods of preparation and use were largely unspecified, or superficially described (again, in sections other than Methods).
3) Data depictions are primarily complex, multi-panel (and somewhat exhausting) photomicrographs. To a large degree the nature of these experiments mandates same, but is there another way to convey the same info, at least on some occasions and thereby break up the repetition?
4) The discussion could be cut by 25-50%. Too much of it is a historical discourse, which to some degree is a reiteration of information in the Introduction. The many examples/details of viral replication were interesting, but probably not all were necessary in the present context. One very protracted paragraph was >>a page in length. More focus on interpreting the present data, and succinct discussion in the context of related/previous findings, would be easier for readers to digest and, consequently, more informative.
Author Response
Comments and Suggestions for Authors
This is a comprehensive, if not exhaustive, demonstration that oxidative stress drives La/SSB nuclear/cytoplasmic translocation. For the most part, the methods are generally sound. This represents an incremental advance from previous work.
Response to reviewer:
We are grateful for this kind evaluation of our work.
Criticisms:
1) The presentation of data is entirely and rather cursorily descriptive; more like a narrative than the typical scientific report.
There are no indications of how many times the experiments were repeated ("n'), data distributions (error bars) nor any attempt at analyzing intergroup differences. The absence of knowing what the data really shows (n, and dispersion) limits the credibility of these findings.
Response to reviewer:
Already in the original submission we mentioned in the statistical section that each experiment was performed at least in triplicates and that the deviation between the experiments were less or around 5%.
2) Many methods were either inadequately detailed and/or primarily presented in figure legends, and sometimes results, rather than in Methods section per se. In particular, the sources of oxidants, TLR ligands, and methods of preparation and use were largely unspecified, or superficially described (again, in sections other than Methods).
Response to reviewer:
We apologize for having not included all information related to the origin of some of the materials and the assays. And, indeed, we had added information to some materials and methods either directly to the figure legend or the result section. We thought it might be easier to follow the respective in part complex experiments if we do so. As requested by the reviewer we now have removed these material and methods related information from the figure legends and the result section. We have included all these information to the material and methods section which therefore was heavily revised and extended. As requested we have now also included the sources of the TLR ligands (LPS, Resiquimod) and other chemicals such as BSO as well. In order to facilitate the reproducibility of our data we have also given the details of the assays including concentrations and exposure times related to: How we oxidized or reduced La protein at the cellular level prior or after fixation of the cells. All these details are now included in the material and methods section.
3) Data depictions are primarily complex, multi-panel (and somewhat exhausting) photomicrographs. To a large degree the nature of these experiments mandates same, but is there another way to convey the same info, at least on some occasions and thereby break up the repetition?
Response to reviewer:
We apologize for the complexity and “somewhat exhausting” photomicrographs. However, all the individual images are required for the whole picture as also accepted and acknowledged by the first reviewer. There have been so many controversial approaches reported that we think it is justified to explain in a very detailed way how the shuttling can be induced and followed and which pitfalls can occur which can lead to failures. And these are minor things such as using methanol for fixation that has been stored under the wrong conditions which may or may not cause problems in combination with anti-La abs recognizing certain redox sensitive epitopes. The same is true for cell culture conditions: Simply the concentration of reducing agents in the media, which will also be influenced by the duration of the culture the frequency of media change etc. Therefore all the details are required which may sound “somewhat exhausting”. However, the importance of these details become evident for example from the results with the endothelial cells in which the shuttling cannot be triggered as long as too high concentrations of reducing agents exist intracellularely. Overall we hope that our detailed summary helps to find solutions to solve problems when not being able to trigger the shuttling rather than the comment the shuttling doesn’t happen because we cannot see it.
4) The discussion could be cut by 25-50%. Too much of it is a historical discourse, which to some degree is a reiteration of information in the Introduction. The many examples/details of viral replication were interesting, but probably not all were necessary in the present context. One very protracted paragraph was >>a page in length. More focus on interpreting the present data, and succinct discussion in the context of related/previous findings, would be easier for readers to digest and, consequently, more informative.
Response to reviewer:
Well discussion/interpretation is usually a personal view. Reviewer 1 was obviously very satisfied with our way to do it. Nonetheless, we agree with the reviewer 2 that we can (and have) intensely shorten(ed) the Discussion section to avoid iterations. E.g. the summary of the function of La protein in translation of cellular and viral mRNAs which was already summarized in the introduction section was completely deleted from the discussion. Thereby, as requested, the discussion automatically is more focused on the presented data.
